# Biomimic Zwitterionic Polyamide/Polyurethane Elastic Blending Fabrics

**DOI:** 10.3390/biomimetics8020198

**Published:** 2023-05-10

**Authors:** Ying-Nien Chou, I-Hsun Yang

**Affiliations:** Department of Chemical and Materials Engineering, Southern Taiwan University of Science and Technology, Tainan 71005, Taiwan

**Keywords:** zwitterionic copolymer, surface modification, anti-biofouling, polyamide fabrics

## Abstract

This study developed an epoxy-type biomimic zwitterionic copolymer, poly(glycidyl methacrylate) (PGMA)-poly(sulfobetaine acrylamide) (SBAA) (poly(GMA-co-SBAA)), to modify the surface of polyamide elastic fabric using a hydroxylated pretreatment zwitterionic copolymer and dip-coating method. X-ray photoelectron spectroscopy and Fourier transform infrared spectroscopy confirmed successful grafting, while scanning electron microscopy revealed changes in the surface pattern. Optimization of coating conditions included controlling reaction temperature, solid concentration, molar ratio, and base catalysis. The modified fabric exhibited good biocompatibility and anti-biofouling performance, as evidenced by contact angle measurements and evaluation of protein adsorption, blood cell, and bacterial attachment. This simple, cost-effective zwitterionic modification technology has high commercial value and is a promising approach for surface modification of biomedical materials.

## 1. Introduction

Biomedical materials can be divided into four major parts: polymers, ceramics, metals, and composite materials. Materials replacing part of ceramics or metals with polymers have become popular in the market. Whether it is synthetic fibers or natural fibers, they are used in daily life. It can be seen everywhere in life, such as the most commonly used masks and surgical gowns in medical treatment, but these generally have no biomedical application capabilities but have the characteristics of easy processing, high toughness, and drug resistance. With the development of functional materials, more and more people pay attention to the application of fibers in the field of biomedicine [1,2,3]. Biocompatibility is defined as implanted biomedical materials without adverse reactions with living organisms, such as inflammation, fever, rejection, blood coagulation, etc. The connection between all materials and living organisms will be different from the chemical or physical properties of the material and leads to various reactions in the living body. Biocompatibility includes blood compatibility and tissue compatibility. Histocompatibility mainly means that the mechanical strength or shape of the material itself must meet the needs of the patient, and the appearance must match the fit of the patient’s wound to avoid adverse phenomena between the tissue and the implant. Blood compatibility mainly prevents foreign biomedical materials from entering the living body and contacting blood to cause coagulation or thrombus. Factors affecting blood compatibility include material surface charge, surface hydrophilicity, hydrophobicity, surface roughness, etc. Most biomedical materials will come into contact with blood. In order to prevent biomolecules in the blood, such as proteins, platelets, and red blood cells, from biofouling, materials must have good biocompatibility and anti-adhesion properties [4].

Biofouling is the contamination of surfaces with microorganisms, including bacteria, fungi, and viruses. In medical applications, biofouling occurs on surgical equipment, protective clothing, packaging, guide wires, sensors, prosthetic devices, and medical implants, most commonly catheters, drug delivery devices, and contact lenses. Microbial contamination causes infection risk, biosensor failure, and implant rejection, and how to avoid biomolecules fouling onto materials is the current trend [5]. According to a study sponsored by the U.S. Centers for Disease Control and Prevention, approximately 26% of infections in U.S. emergency hospitals alone in 2011 were caused by these device-associated infections [6]. Since biofouling is protein-mediated, inhibition of protein adsorption can prevent infection at the source, with the surface-adsorbed protein layer serving as a platform for cell attachment and subsequent bacterial film formation [7]. Protein contamination is a significant challenge in the development of many blood-contacting biomedical devices due to the nonspecific adhesion of biological components, including proteins, to the device’s surface. When proteins are adsorbed to a surface, a cascade of biological reaction, these processes lead to thrombosis [8], which in turn leads to platelet formation and ultimately to device failure and fatal complications. Additionally, the adhesion of bacteria to surfaces is mediated by different types of interactions, which can be specific, such as through protein films that may form on the surface or through nonspecific interactions [9,10], attached to the surface to form a biofilm [11]. Biofilm formation on biological implants such as catheters, prosthetic devices, and contact lenses can lead to infection. Typical treatments for biofilm-induced infections on medical devices include surgical replacement of contaminated devices and prolonged antibiotic treatment, which creates additional healthcare costs. Anti-biofouling materials are currently used in many applications, such as marine, industrial and biomedical applications [12].

The technical source of hydrophilic anti-bioadhesive materials is that by forming a hydrophilic structure on the surface of the material, a hydration layer can be created, thereby preventing proteins from hydrophobic adsorption. Currently, known hydrophilic anti-bioadhesive materials are divided into three generations. Namely, the first-generation 2-hydroxyethyl methacrylate (HEMA-based system) with the structure of -OH, the second-generation ethylene glycol (PEGlyated system) with a (C-C-O) structure, and the third-generation zwitterionic material (Zwitterinoic system) has an electrically neutral structure with both positive and negative charges in the same chain segment [3,13]. Among them, zwitterionic polymers have a strong hydration ability and are considered to be the best anti-biofouling material solution. Common ones include phosphorylcholine (PC), carboxybetaine (CB), and sulfbetaine (SB). Compared with PC and CB, polySBMA (poly sulfbetaine methacrylate) polymethacrylate sulfobetaine and polySBAA (poly sulfbetaine methacrylate) polymethacrylamide sulfobetaine composed of sulfobetaine sulfbetaine (SB) The advantages of simple and low-cost synthetic steps make it the most widely studied zwitterionic polymer [14,15]. To achieve good anti-biofouling properties and industrial applicability, simple and stable surface modification of biomaterials with zwitterionic polymers is crucial. Among them, epoxy-based zwitterionic copolymers have been designed to enable simple and extensive surface modification of various types of substrates, such as metals, ceramics, and polymers, and can form a stable anti-biofouling modification layer through chemical bonding on the surface [16]. In recent years, epoxy-based zwitterionic copolymers have been reported for modifying and applying various types of substrates, such as PET [17], titanium/stainless steel [18], chitosan [19], PP [20], PDMS [21], and PTFE [22].

Polyamide (Nylon), which is a synthetic fiber, was invented in 1935. Compared with natural fibers, it has a high yield, high mechanical properties, corrosion resistance, weather resistance, and low cost [23]. The utilization rate of nylon fibers is increasing year by year. For textile applications, with the rapid development of science and technology, there are more and more applications of nylon, such as the application of biosensors [24,25]. Still, although nylon itself has certain biocompatibility and biological inertness, it is not enough. Resisting the adhesion of biomolecules causes the sensitivity to decrease, mainly due to the lack of hydrophilic and high coverage factors. In early 2001, plasma was used to modify the surface to improve the hydrophilicity of materials [26]. There are three main types of plasma-modified surfaces, namely Environmental discharge [27], microwave method, and ion beam discharge [28], although these methods can improve the hydrophilicity and the surface charge can be anti-bacterial, but the stability could be better.

Medicine science is one of the most innovative and developing aspects of the textile industry; with the development of the field of biomedical applications, recent years have witnessed the production of medical textiles, using materials including monofilament, multifilament, woven, knitted, and non-woven cloth, due to the wide demand, medical textiles occupy a huge market, and the use of natural fibers and synthetic fibers in the production of various medical products has increased sharply. According to the research of DRA (David Rigby Associates), the global medical and hygiene products in 2000, more than 1.5 million tons of textile materials, worth USD 5.4 billion, were consumed in the manufacture of textiles. By 2010, the quantity had increased by 4.5% annually to 2.4 million tons, worth USD 8.2 billion. It can be seen that there is a huge economic effect behind it [29]. Among them, polyamide elastic fabrics have been widely developed, spanning from clothing to biomedical materials, but they have fatal shortcomings in clinical medical applications, that is, insufficient biocompatibility, and cannot resist the adhesion of biomolecules in the body, resulting in abnormal immune systems. However, the surface of polyamide elastic fabric is relatively inert. According to previous research [25], using atom transfer radical polymerization (ATRP) to graft zwitterionic carboxybetaine methacrylate (CBMA) on its surface can successfully reduce the amount of protein adsorption by about 65%. The advantage of this surface starting method (grafting from) is that the surface polymer brushes can be arranged neatly and densely to achieve a better anti-biofouling effect. The disadvantage is that the steps are complicated, the cost is high, and it is difficult to industrialize mass production.

In a previous study, we learned that epoxy-based zwitterionic copolymers could form a good and stable anti-biofouling modification layer with surface hydroxyl groups (-OH) [16]. Therefore, this study designed a new type of copolymer structure, the epoxy-based zwitterionic acrylamide copolymer, poly(GMA-co-SBAA), was fixed on the polyamide composed of nylon 6,6 and polyurethane by surface grafting method (grafting to). The elastic fabric surface makes the surface electrically neutral and electrostatically interacts with water molecules, forming a hydration layer on the surface to resist the adhesion of biomolecules, as shown in Figure 1. In order to create a stable and sufficient hydroxyl group (-OH) on the surface of the PA fabrics, we refer to the hydroxylation pretreatment process by formaldehyde in the previous study [25]. This process can make the amide structure react with formaldehyde to generate a large amount of hydroxyl group (-OH) and is capable of reacting with the epoxy group in GMA. The modification of poly(GMA-co-SBAA) on the polyamide elastic fabric could create high hydrophilicity and high biocompatibility, which can effectively improve biocompatibility and anti-biofouling properties.

## 2. Materials and Methods

### 2.1. Materials

Glycidyl methacrylate (GMA), triethylamine (TEA), ammonium persulfate (APS), phosphate-buffered saline (PBS), acetic acid, methanol (MeOH), ethanol, acetone, formaldehyde, tetrabutylammoium hydrogen sulfate, glutaaldehyde 50 wt.% solution in water, bovine serum albumin (BSA), human fibrinogen, ampicillin sodium salt (AMP), and LB Medium, were purchased from Sigma-Aldrich Chemical Co. Sulfobetaine acrylamide (SBAA) was purchased from Hopax Chemicals Co. (Kaohsiung, Taiwan). The bicinchoninic acid (BCA), 2-hydroxyethyl methacrylate (HEMA), diiodomethane were purchased from Alfa Aesar Chemical Co. (Kaohsiung, Taiwan). Polyamide elastic fabrics (Nylon 6,6, 40D, 10% Spandex) were provided from Any Color International Limited Co. (Tainan, Taiwan). Deionized water (DI water) was purified using the Millipore water purification system with a minimum resistivity of 18.2 MΩ ∙ cm.

### 2.2. Synthesis and Structure Identification of Zwitterionic Poly(GMA-co-SBAA) Copolymer

Dissolve the monomers SBAA (sulfobetaine acrylamide) and GMA (glycidyl methacrylate) in 25 mL aqueous solution and 25 mL methanol solution, respectively, with a solid content of 20%, and add the two bottles of solutions into a 100 mL reaction bottle (Molar Ratio: SBAA:GMA:APS = 40:60:1). After thoroughly stirring the solution, it should be filled with nitrogen gas for 10 min and placed in a silicon oil pot at 60 °C for 6 h. Then, the reaction bottle should be transferred to an ice bath at 4 °C, and the solution should be poured into methanol for purification. The resulting polymers should be dried in a vacuum oven and then treated with a freeze-dryer to remove any remaining water. Finally, the polymers should be stored in a vacuum-sealed container to ensure their stability. In this experiment, a nuclear magnetic resonance (NMR) spectrometer (Bruker, NMR-400 MHz, Billerica, MA, USA) with a resonant frequency of 400 MHz was used to detect the chemical structures of monomers and zwitterionic copolymers. Monomer GMA detection: Dissolve 20 mg GMA monomer in 0.5 mL d_6_-DMSO, and put the detection solution into an NMR tube to detect hydrogen spectrum. Monomer SBAA detection: Dissolve 3.5 mg of SBAA monomer in 1.0 mL of D_2_O, and put the detection solution into the NMR tube to detect the hydrogen spectrum. Zwitterionic polymer detection: Take 3.5 mg polymer powder and dissolve it in 1.0 mL D_2_O, oscillate with Vortex to dissolve the polymer as much as possible, put the solution into the NMR tube, and detect the hydrogen spectrum.

### 2.3. Surface Modification and Identification of Hydroxylated Pretreated Zwitterionic Copolymers on Polyamide Elastic Fabrics

Polyamide plates are hot-pressed at 280 °C through nylon 6,6 ester grains with a double-action hydraulic molding machine to form rectangular blocks of 2 × 4 × 0.1 cm^3^ and then cut into 1 × 1 × 0.1 cm^3^ square block by scissors and keep it in a vacuum ball for use. The polyamide elastic fabrics used in this study are a blend of 90% polyamide (Nylon 6,6) and 10% polyurethane (PU, Spandex). The polyamide elastic fabric was cut into 1 cm^2^ squares, put into a formaldehyde solution with phosphoric acid as a catalyst, reacted for 12 h, and repeatedly rinsed with DI water more than 5 times to obtain a polyamide elastic fabric with hydroxyl groups on the surface [25]. Notably, usage of formaldehyde is toxic, and appropriate safety measures should be taken during operation. The experiments must be conducted in a well-ventilated fume hood. Then the polymer poly(GMA-co-SBAA) was prepared into a polymer solution with water as a solvent. Triethylamine was used as a catalyst to put the cut nylon fiber cloth into the solution and placed at a speed of 110 rpm and a temperature of 60 °C for 9 h in an oven. Finally, the modified polyamide elastic fabric was shaken with ultrasonic waves (DC 150H, Delta Co., Tainan, Taiwan) for 20 min to shake off the poorly connected polymers, dried in a freeze dryer, and stored in a vacuum ball.

### 2.4. Identification of Surface Physical and Chemical Properties of Polyamide Elastic Fabric

The types of functional groups on the surface of the material were identified using Fourier Transform Infrared Spectroscopy (FT-IR) (Perkin Ellmer Auto/MAGE FT-IR Microscope). The physical structure of the surface was observed using a scanning electron microscope (SEM, Phenom ProX, Thermo Fisher Scientific, Waltham, MA, USA) at an accelerating voltage of 15 KV. First, the sample was pasted on a particular stage of a scanning electron microscope with carbon tape and put under vacuum conditions for 24 h. Before transfer into the SEM chamber, the sample was deposited with a gold layer for 90–100 s by the gold deposition machine to increase the electrical conductivity. X-ray photoelectron spectroscopy (PHI 5000 VersaProbe Ⅲ) measurements were conducted using a PHI Quantera SXM/Auger spectrometer with a monochromatic Al KR X-ray source (1486.6 eV photons), a hemispherical analyzer, and a multichannel detector. The elemental signal was determined from spectra acquired at pass energy from 50 eV to 150 eV. The detected photoelectron was acquired at the takeoff angle of 45 with respect to the sample surface. All the binding energy was referenced to the C1 spectrum at 284.4 eV. In order to identify the polymer grafting density on the surface of the substrate, the polyamide elastic fabric was first cut into a square with a side length of 1 cm by an infrared cutting machine (SUII-4060, BAXCE technology, Tainan, Taiwan) and put into 24 Well plates, one piece per grid, and placed in the freezer. Remove the water on the surface in the drying machine, then take each piece of polyamide elastic fabric to a microbalance and weigh it (microbalance AP135W, SHIMADZU, Kyoto, Japan) to obtain the weight before modification W_1_, then take each part of polyamide elastic fabric to modify, and then remove the modified The finished polyamide elastic fabric is placed in a freeze dryer (FD-4.5-12P, Ker-shi tech., Tainan, Taiwan) to remove the water on the surface, and then weighed with a microbalance to obtain the modified weight W_2_.
Hydration capacity (mg/cm^3^) = (W_2_ − W_1_)/V

W_1_: membrane weight before modification (mg)W_2_: membrane weight after modification (mg)V: membrane volume (cm^3^)

Because of the high porosity and original hydrophilic property of polyamide fabrics, surface static or dynamic water contact angle can not sufficiently be detected. Thus, the surface hydrophilicity of polyamide elastic fabrics is measured by the oil contact angle by an angle meter (FTA 125 Contact angle analyzer, Newark, CA, USA). During the measurement, a quantitative 4 μL of diiodomethane is dropped on the surface of the nylon fiber membrane and then measured by a contact angle meter. The included angle between the droplet and the sample surface is used to judge the hydrophilicity of the samples.

### 2.5. Moisture Content Test of Polyamide Elastic Fabric

Cut the polyamide elastic fabric into squares with a diameter of 1 cm using an infrared cutting machine, put them into 24 Well plates, put one piece in each grid, place them in a freeze dryer to remove the surface moisture, and then put each piece Take the polyamide elastic fabric to a microbalance and weigh it to obtain the dry weight W_1_, then take each piece of polyamide elastic fabric to modify, soak it in 1 mL of DI water after modification, and put it in 37 °C Oven for 12 h, so that the polyamide elastic fabric can be evenly wetted, blot the excess water on its surface with lens tissue, and then weigh it with a microbalance to obtain the wet W_2_.
Hydration capacity (mg/cm^3^) = (W_2_ − W_1_)/V

W_1_: membrane weight before wetting (mg)W_2_: film weight after wetting (mg)V: membrane volume (cm^3^)

### 2.6. Anti-Biofouling Test of Polyamide Elastic Fabric Surface

In the protein adsorption experiment, bovine serum albumin (BSA) was adsorbed, and BCA Protein Assay Kit was used for color development to quantify the amount of protein adsorbed on the surface. The modified 1 cm^2^ polyamide elastic fabric was placed in a 24-well plate. Add 1 mL of PBS to each well of polyamide elastic fabric, put it in a 37 °C incubator, and place it for 10–12 h to ensure that the polymer on the fabric is stretched and thoroughly moistened. Remove the PBS in the 24 well plates, and then rinse with PBS 3 times to wash away the polymers that have not been successfully modified on the polyamide elastic fabric. Add 1 mL of 1 mg/mL PBS bovine serum albumin solution to the polyamide elastic fabric, and put it in a 37 °C incubator for 30 min to absorb. The bovine serum albumin was removed, and the polyamide elastic was rinsed three times with PBS. Move the polyamide elastic fabric from the original 24-well plate to the new 24-well plate to avoid the reaction of bovine serum albumin on the original well plate and affect the calculation of the adsorption capacity. Add 1 mL of BCA chromogenic solution to a new 24-well plate, and put it in a 37 °C incubator for 30 min. After the reaction is over, pipette 200 μL of the liquid into the 96 well plates. Using a micro-disk spectrometer to measure the absorbance with a wavelength of 562 nm using a micro-disk spectrometer (SPECTROstar Nano, BMG LABTECH, Ortenberg, Germany) and convert the adsorption amount of bovine serum albumin.

The blood adhesion test is performed in an IRB-certified laboratory. The modified 1 cm^2^ polyamide elastic fabric is placed in a 24-well plate, and 1 mL of PBS is added to each piece of polyamide elastic fabric in each well. It was then placed in a 37 °C incubator and soaked overnight; we removed the PBS and then rinsed it with PBS 3 times and added blood (whole blood is whole blood, RBC is red blood cell thick solution) to the polyamide at 1 mL per cell. The elastic fabric was then put in a 37 °C incubator and stuck for 1 h; we then removed the blood and rinsed it with PBS three times. In the cases, add 1 mL of glutaraldehyde at a concentration of 2.5% to each piece of nylon fiber to fix the cells, and soak for 3 h, then remove the glutaraldehyde and rinse with PBS three times, place it in a cool place to dry naturally, and a scanning electron microscope was used to observe the sample. Finally, the software Image J was used to calculate the number of attached blood cells in the blood.

For the bacterial attachment experiment, first, add 12.5 g LB Medium and 500 mL DI water to a 500 mL serum bottle, stir at room temperature until dissolved, cover the lid of the serum bottle with aluminum foil bright bread, paste sterilizing tape, and place in a sterilizing kettle. After 20 min of wet sterilization, put it into a biological safety cabinet, add 500 μL AMP after the temperature cools down to 37–45 °C, and store it in an oven at 37 °C. The strain used is *Escherichia coli* (*E. coli*, G), Gram-negative bacterium, the surface is negatively charged at pH = 7.4, and the concentration of the culture solution reaches 8 × 10^8^ cells/mL. Take 60 mL of culture solution and put it into T-75 flask, then add 0.6 mL of fresh solution, and place it in an incubator at 37 °C and 100 rpm for 16–18 h. Place the modified 1 cm^2^ polyamide elastic fabric in a 24-well plate, add 1 mL of PBS to each piece of polyamide elastic fabric in each well, put it in a 37 °C incubator, and place it for 10–12 h, remove the PBS and rinse 3 times, then add 1 mL cultured solution to each cell, and put it in a 37 °C incubator, wait for 1 h to draw out the bacterial solution, wash 3 times with PBS to remove unattached bacteria or impurities. Add 2.5 vol% glutaraldehyde solution, place in a refrigerator at 4 °C for 3 h, then remove glutaraldehyde and rinse with PBS three times, place in a cool place to dry naturally, and then use a scanning electron microscope to carry out sample observation. Finally, use Image J to calculate the amount of blood attached.

## 3. Results and Discussion

### 3.1. Identification of Zwitterionic Copolymer Polymers

In this study, monomeric structure sulfobetaine acrylamide (SBAA) was copolymerized with glycidyl methacrylate GMA to form poly(GMA-co-SBAA) zwitterionic copolymer. Compared with the poly(GMA-co-SBMA) used in previous studies [16], Yung Chang’s team recently reported that the SBAA structure could have the same good biocompatibility and better vapor thermal stability [30]. In addition, the amide bond of SBAA is expected to generate more intermolecular forces on the modification of polyamide substrates, thereby achieving better modification results. In addition, because the amide bond of SBAA is easier to generate intermolecular hydrogen bonds than SBMA, it can be observed during synthesis that the molecular weight tends to rise rapidly with the polymerization time and the increase in polymerization temperature. When the polymerization is terminated, a precipitate will form a slightly transparent state colloids. The polymerization of polymers containing SBAA must be properly and accurately controlled for polymerization time and temperature to obtain better polymer properties.

Figure 2 shows whether poly(GMA-co-SBAA) was successfully synthesized by proton nuclear magnetic resonance (^1^H-NMR). The signals appearing at δ = 2.63 ppm, δ = 2.84 ppm, and 3.5 ppm correspond to the characteristic peaks of the epoxy groups of methylene (l) and methine (k), whose protons are from PGMA. At the same time, the proton signals of the methylene(j) closest to the epoxy group appeared at δ = 3.8 ppm and δ = 4.5 ppm [31]. In addition, δ = 3.2 ppm, δ = 5.5 ppm, and δ = 5.8 ppm correspond to the proton resonance and amine group (g) of (CH_3_)_2_N^+^ (i) on SBAA, respectively [32], thus proving the successful synthesis of poly (GMA-co-SBAA).

Many previous studies have pointed out that the two-component composition of copolymers with modification or coating properties plays an important role in the modification properties [16]. In order to explore the modification properties of polyamide substrates in the future, Poly(GMA-co-SBAA) with different dual monomer compositions will be synthesized by adjusting the molar ratio of the mixed reaction of GMA and SBAA, according to the identification of molecular weight (Mw) and molecular weight distribution index (PDI) by colloid permeation chromatography. The synthesis method of poly(GMA-co-SBAA) is free radical polymerization in solution polymerization. Its molecular weight is about 30 kDa to 46 kDa, and its PDI value is about 1.5 to 2, which means that the molecular weight distribution of the polymer is relatively dense. In addition, the length of the SBAA chain segment will affect the amount of protein adsorption and improve the level of anti-adhesion performance. The length of the GMA chain segment will affect the surface grafting density. To achieve the best anti-adhesion effect, we synthesized various poly(GMA-co-SBAA) compositions, as shown in Table 1, poly(GMA-co-SBAA) zwitterionic copolymer’s theory and actual molar ratio, molecular weight size, and molecular weight distribution index.

### 3.2. Surface Modification of Zwitterionic Copolymer by Hydroxylation Pretreatment

Achieving the zwitterionic anti-biofouling polyamide blending fabrics requires careful process optimization, including the selection of copolymer configuration and chain length, as well as the optimization of the dip-coating process parameters, as shown in Figure 3a. The incorporation of zwitterionic groups into the polyamide blending fabrics can significantly enhance their anti-biofouling properties by reducing protein adsorption and bacterial attachment. The optimization of copolymer configuration and chain length can enhance the surface hydrophilicity and protein resistance, while the optimization of coating concentration, coating time, and additive catalysts can ensure a uniform and stable coating of the copolymer on the surface of the polyamide blending fabrics, which can enhance the anti-biofouling properties. When the polyamide elastic fabric is modified with different molar ratios between PGMA and PSBAA, the amount of bovine serum albumin adsorbed by poly(GMA-co-SBAA) changes to explore which ratio can have the best anti-adhesion effect. In this study, a slight amount of formaldehyde was used as a pretreatment to generate sufficient reactive hydroxyl groups. After formaldehyde treatment, the samples were subjected to multiple washes and modified with a biocompatible zwitterionic polymer in an alkaline environment. It can be observed from Figure 3b that the polyamide elastic fabric obtained through pretreatment may increase its hydrophilicity due to the hydroxyl group (-OH), which improves its protein resistance, and it can be found that when the proportion of GMA increases, due to the increase in graft density. When the ratio of poly(GMA-co-SBAA) is G70S30, the anti-adhesion effect begins to decrease, which is due to the lack of zwitterionic groups with anti-adhesion properties. From Figure 3b, it can be found that the monomer ratio of the copolymer can achieve the best modification performance in G60S40. Interestingly, compared with previous optimizations of epoxylated zwitterionic copolymers [16], the optimal copolymer composition ratio found in this study is not 20% for the epoxy group and 80% for the SB group. This may be due to the differences in the substrates. PA/PU fabrics cannot contain a large amount of -OH groups after the formaldehyde treatment, so more epoxy groups are required in the copolymer to have sufficient driving force when performing alkali-catalyzed surface modification. We carried out a sufficient surface ring-opening bonding reaction with PA/PU fabric substrates.

After finding the optimal molar ratio of poly(GMA-co-SBAA) to be G60-S40, modify the polyamide elastic fabric by the copolymerized polymer obtained by PGMA and PSBAA at different copolymer monomer and initiator ratios. According to past experience, the greater the ratio of monomer to the initiator, the higher the molecular weight of the copolymer can be obtained, and the change in the amount of bovine serum albumin adsorbed by the polyamide elastic fabric is used to explore the optimal molecular weight. The anti-adhesion effect of G60-S40 is the best anti-adhesion effect, as shown in Figure 3c.

Figure 3d presents Poly(GMA-co-SBAA) (G60-S40) modified polyamide elastic fabrics under different catalysts to explore the optimal catalyst through the change of the amount of bovine serum albumin adsorbed on the surface. According to the previous study, it can be seen that when the surface has hydroxyl groups, it has a good reactivity with the GMA system under the alkaline environment of adding triethylamine. It is observed that alkaline catalysis has better anti-fouling performance than acid catalysis, and the reason why glacial acetic acid is better than amine water may be that there are unreacted amine groups that are easier to expose and react with GMA under acidic conditions. The overall reactivity is better than amine water. Then use the above optimal conditions to explore the modification time and the change of bovine serum albumin adsorption of modified polyamide elastic fabrics of G60-S40 at different concentrations. It is observed from Figure 3e that when the reaction time is 9 h, it is the optimal reaction time. However, when the reaction time reaches 12 h, the overall anti-adhesion performance decreases due to the accumulation of polymers on the surface or too serious agglomeration. In contrast, the concentration in Figure 3f also showed the same trend. When the concentration was increased from 5 mg/mL to 10 mg/mL, the above two possibilities can also be found. Through the discussion of the above optimization parameters, the conclusion is: when the poly(GMA-co-SBAA) mol ratio is G60-S40, the molecular weight is G60-S40 (100:1), the catalyst is triethylamine, and the polymer concentration is 5 mg/mL and the modification time of 9 h are the best modification parameters.

### 3.3. Surface Modification of Polyamide Blocks Made of Nylon 6,6 and Polyurethane

The polyamide (PA) elastic fabric used for modification in this study is composed of nylon and polyurethane mixed yarns. In order to confirm that the copolymer can be effectively modified on the two substrates, we separately target the flat blocks composed of the two materials. Copolymer modification was performed and tested for resistance to protein adsorption. We use pure polyamide (Nylon 6,6) and polyurethane flat plate with high structural similarity as the modified substrate to ensure that chemical grafting can be formed on a smooth surface and confirm that this method can be successfully modified on porous polyamide elastic fabrics. Poly(GMA-co-SBAA) is used to modify the surface of smooth surface polyamide plate (Nylon 6,6) and polyurethane (Polyurethane, PU) plate with hydroxylated pretreated zwitterionic copolymers, and after modification, ultrasonic shake for 20 min to remove polymers with poor bonding strength, and use water contact angle and protein adsorption experiments to verify that the surface has improved hydrophilicity and has a good anti-biofouling effect.

As shown in Figure 4, after modification, the PA plate can form an electrostatic force with water because of the zwitterionic structure on the surface, resulting in the formation of a hydration layer on the surface and increasing its hydrophilic properties. The experimental results showed that the water contact angle decreased significantly, and the protein adsorption decreased by about 98%, which confirmed that the improvement of hydrophilicity was positively correlated with the resistance to protein adhesion.

About five million venous PU catheters are used in patients every year in the United States. If such catheters are not properly handled, they are susceptible to bacterial adhesion and biofilm formation, and they are easily used in the body to cause blood-borne infections [33]. Therefore, we use poly(GMA-co-SBAA) functional polymer copolymer to modify PU blocks to make them have surface anti-biofouling properties, which are used to resist bacterial adhesion and avoid biofilm formation. Figure 4 shows that compared with the unmodified PU plate, the water contact angle of the modified PU is significantly reduced, and the increase in hydrophilicity is also positively correlated with the resistance to protein adhesion.

### 3.4. Surface Characterization through Chemical Analysis of FTIR-ATR and XPS

In order to ensure the successful modification of polyamide elastic fabrics, the changes of surface functional groups of polyamide elastic fabrics before and after modification were identified by Fourier transform infrared spectroscopy (FT-IR/ATR). Poly(GMA-co-SBAA) contains C = O functional group, its wavelength falls between 1714–1746 cm^−1^, and the characteristic peak of SO_3_^−^ is 1035 cm^−1^ [34]. It can be observed from Figure 5 that after modification by copolymer G60-S40, the characteristic peaks of zwitterion with SO_3_^−^ have obvious changes, which can confirm that the grafting rate of the surface polymer surface has indeed increased, proving that this method is successfully modified in polyamide elastic fabric.

X-ray photoelectron spectroscopy (XPS) was used to analyze the changes in the surface elements of the polyamide elastic fabric before and after modification. The polyamide elastic fabric itself contains C, N, and O elements, while the zwitterionic SBAA contains C, N, O, and S elements, so the change of N and S elements can be used to investigate whether the zwitterionic polymer is successfully modified. From Figure 6, in the N1s part, after modification by copolymer, the spectrum contains two peaks representing the structure N-H and quaternary ammonium cation NH_4_^+^, with binding energies of 399.2 eV and 402.8 eV, respectively. In the S2p part, the spectrum also provided further evidence for the existence of the zwitterionic part, as a clear peak was detected in the spectrum with a binding energy of about 167.2 eV, representing the presence of the sulfonic acid functional group of SBAA [35]. In addition, it is also discussed whether the reaction will be affected if no catalyst is added. It is found that there is no change in the N and S signals without adding a catalyst. It is speculated that the epoxy group cannot be successfully ring-opened and grafted without a catalyst onto pretreated polyamide elastic fabric. Table 2 shows the quantitative data of nitrogen and sulfur content before and after the modification of polyamide elastic fabric.

### 3.5. Analysis of Physical Surface Structure of Polyamide Elastic Fabrics

Figure 7 shows the surface morphology of the polyamide elastic fabric before and after modification, and the surface morphology of the unmodified polyamide elastic fabric is a smooth and flat fiber surface, as shown in Figure 7a. In Figure 7, for different molar ratios G20S80, G40S60, G60S40, and G80S20, the surface graft density changes were discussed. After modification, white clusters appeared on the surface of the polyamide elastic fabric, which was speculated to be a copolymer poly(GMA-co-SBAA), which are aggregation patterns in a dry state. After modification by the copolymer poly(GMA-co-SBAA), as the ratio of GMA increases, clearer surface clusters appear, and the grafting density of the polymer surface increases significantly. In addition, Figure 7c shows the changes before and after concentration modification. It is observed that as the concentration increases, the grafting density of polymers on the surface increases. When the concentration increases from 5 mg/mL to 10 mg/mL, the decrease in the surface grafting density may be due to the fact that when the concentration is too high. The macromolecules are seriously agglomerated and cannot be successfully grafted on the polyamide elastic fabric.

### 3.6. Anti-Biofouling Test of Modified Polyamide Elastic Fabric

Figure 8 shows the relative changes in human fibrinogen adsorption and graft density, hydration force, and oil contact angle. The pretreated surface has a large number of hydroxyl groups (-OH). With the increase in the GMA ratio, the surface polymer graft density gradually increases. G80-S20 has the highest graft density, and the moisture content is greatly improved compared with the unmodified polyamide elastic fabric. Although the graft density of G60-S40 is lower than that of G80-S20, the chain length of SBAA is longer. It can capture more water molecules so that the water content is equivalent to G80-S20. Compared with the unmodified polyamide elastic fabric, the water content of G60-S40 is increased from 47 mg/cm^3^ to 82 mg/cm^3^. The oil contact angle increased from 110° to 123°, and the adsorption amount of human fibrin albumin decreased by 74%.

The blood cell adhesion test mainly explores the anti-blood cell adhesion effect of a zwitterionic copolymer modified with different copolymer molar ratio poly(GMA-co-SBAA). In the case of a single red blood cell, it can be seen from Figure 9a that the modified polyamide elastic fabric, compared with the complex whole blood environment, has about 79% less red blood cell attachment, poly(GMA-co-SBAA). The optimal ratio is between G40-S60 and G60-S40. From Figure 9b, it can be concluded that the modified polyamide elastic fabric adheres to the whole blood compared with the unmodified one, and the best-modified result can reduce the adhesion of blood cells by about 75%. However, when the modified polyamide elastic fabric was tested in blood, it was found that the optimal ratio of poly(GMA-co-SBAA) was between G40-S60 and G60-S40, but the trend was not consistent with the fibrinogen adsorption experiment. The results fully correspond to the best anti-biofouling copolymer monomer molar ratio condition in the blood experiment is the polymer composed of G40-S60. Although it does not correspond to the protein adsorption properties, they are all within a close range. It is speculated that during the modification process, due to the conversion of the surface amide bond into hydroxyl (-OH), which is different from the reaction path between the terminal amine group (-NH_2_) and GMA, the steric barriers caused by the surface polymer chains are different, resulting in blood cells are stuck between polymer chains. *Escherichia coli* (*E. coli*), which belongs to Gram-negative bacteria, was selected as the test strain in the bacterial attachment experiment. As shown in Figure 9c, the anti-adhesion results of bacteria tested by pretreatment and modification with zwitterionic copolymers echoed the previous results of protein adsorption resistance. G20-S80 can achieve 68% anti-bacterial adhesion. As the graft density increases, the anti-adhesion effect increases, and the anti-adhesion effect gradually increases after the optimal modification parameters G60-S40, up to 99.2% of the anti-adhesion effect.

## 4. Conclusions

In this study, the zwitterionic copolymer poly(GMA-co-SBAA) was modified on the polyamide elastic fabric by means of surface grafting (grafting to), making it a highly biocompatible and anti-biofouling fabric. The optimal modification parameters are the molar ratio of copolymer monomers of PGMA and PSBAA is 0.6, the concentration is 5 mg/mL, the temperature is 60 °C, the reaction time is 9 h, and the catalyst is triethylamine. In terms of surface property identification, the use of X-ray photoelectron spectroscopy and Fourier transform infrared spectroscopy proved that poly(GMA-co-SBAA) was successfully grafted on the surface of polyamide elastic fabrics and then observed through a scanning electron microscope before and after modification. The research found that the modified polyamide elastic fabric has better biocompatibility and successfully resisted about 70% of human fibrinogen adsorption, 93% of whole blood adhesion, 95% of red blood cell adhesion, and 99% of bacterial adhesion. In addition, the surface modification methods of polyamide elastic fabrics currently include Michael’s addition SI-ATRP. The steps are too complicated and expensive to be industrialized. The poly(GMA-co-SBAA) developed in this research has a facile, convenient, and effective coating method in terms of modification, and this technology has the opportunity to become a continuous process that is applied to biomedical materials and moisture absorption. According to the research on the modification of polyamide elastic fabrics, there are the following important achievements:

Successful modification of polyamide elastic fabric with epoxy-based zwitterionic copolymers so that the material has good blood compatibility and anti-biofouling properties and can resist about 70% of fibrin albumin adsorption and 93% of whole blood adhesion, 95% red blood cell attachment and 99% bacterial attachment.

Surface grafting of poly(GMA-co-SBAA) on the PA/PU fabrics is a convenient and low-cost method for modifying polyamide/polyurethane elastic fabrics.

## Figures and Tables

**Figure 1 biomimetics-08-00198-f001:**
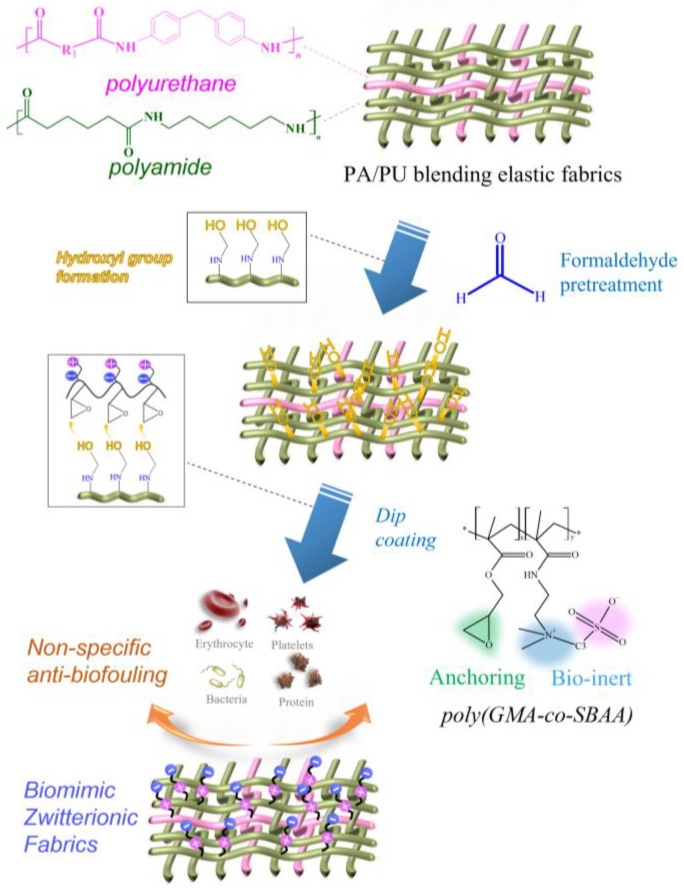
Schematic illustration of the zwitterionic modification of PA/PU fabrics through formaldehyde pretreatment and ethoxylated poly(GMA-co-SBAA).

**Figure 2 biomimetics-08-00198-f002:**
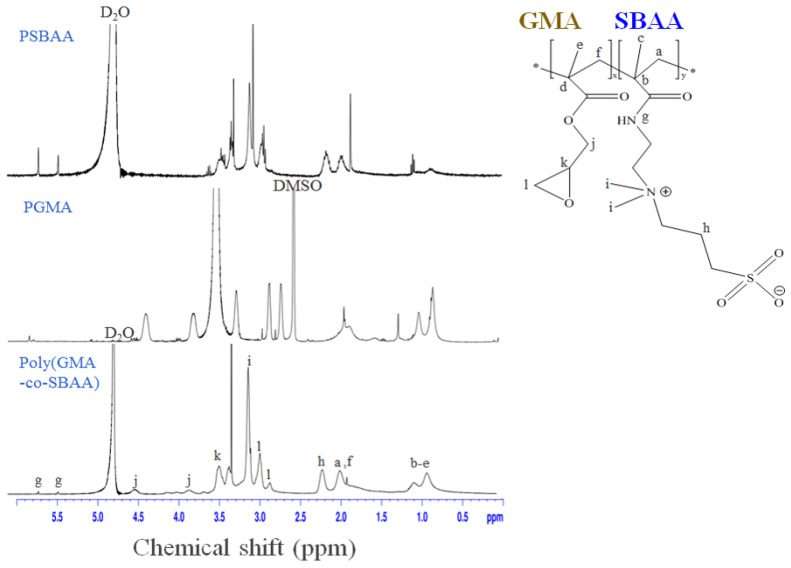
^1^H-NMR diagram of homopolymers, PGMA, PSBAA, and the copolymer, poly(GMA-co-SBAA).

**Figure 3 biomimetics-08-00198-f003:**
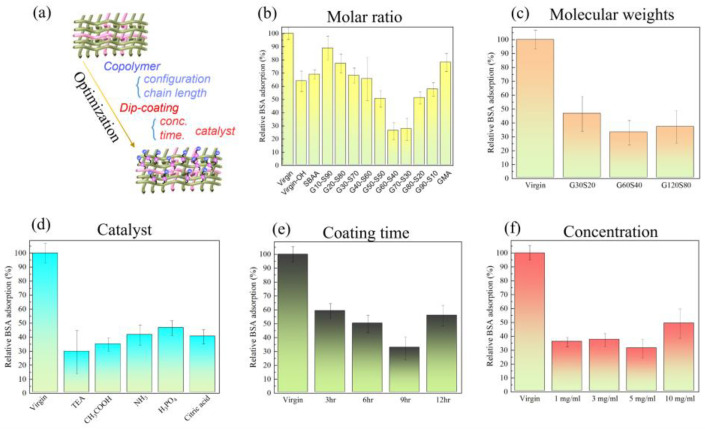
Under the modification of polyamide elastic fabric with poly(GMA-co-SBAA) with different parameters, the change of bovine serum albumin surface adsorption was investigated, (**a**) graphic illustration of the optimization of zwitterionic PA/PU fabrics, (**b**) different molar ratio of monomer, (**c**) different molecular weight of copolymer, (**d**) various acidic and basic catalysts, (**e**) different coating time, and (**f**) different modification dip-coating concentration.

**Figure 4 biomimetics-08-00198-f004:**
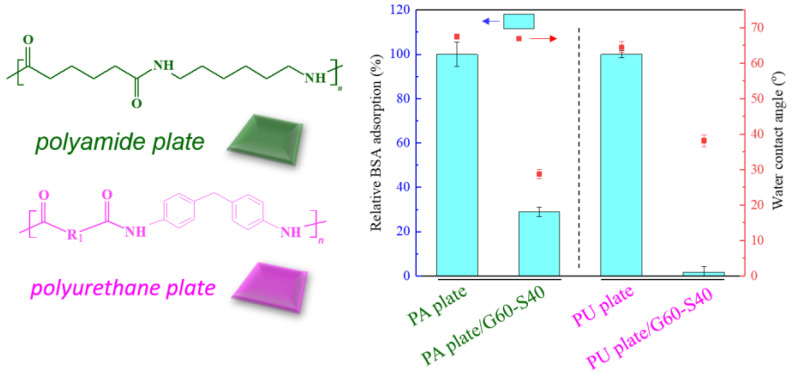
Relative changes of bovine serum albumin adsorption and water contact angle between modified Nylon 6,6 and PU blocks.

**Figure 5 biomimetics-08-00198-f005:**
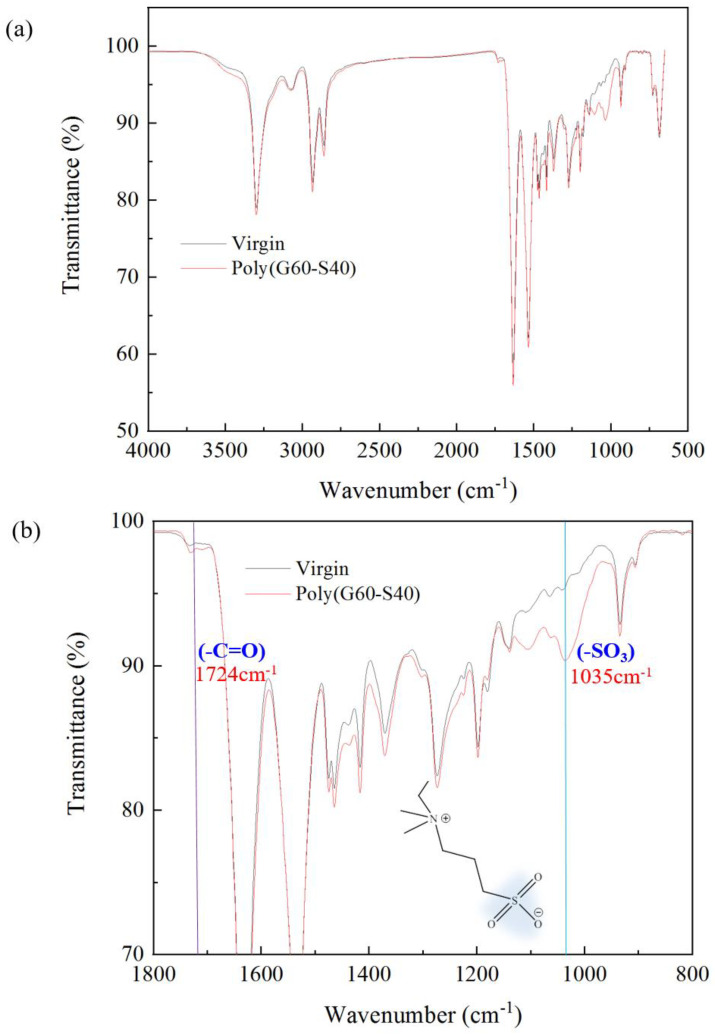
Changes in functional groups of polyamide elastic fabrics before and after modification by Fourier transform infrared spectroscopy (FT−IR/ATR). (**a**) is the full–band spectrum, and (**b**) is the 800–1800 cm^−1^ in (**a**). Zoom in on the spectrum. The sample “virgin” is noted as the original PA/PU fabrics without modification.

**Figure 6 biomimetics-08-00198-f006:**
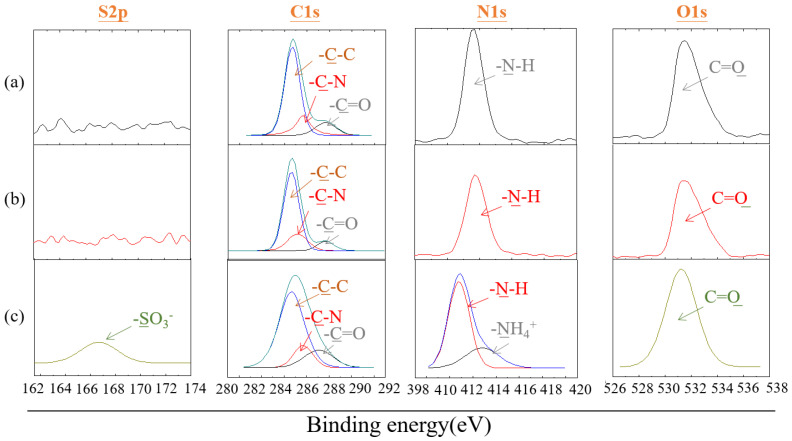
X-ray photoelectron spectroscopy (XPS) patterns of polyamide elastic fabric before and after modification with poly(GMA-co-SBAA) (**a**) original PA/PU fabrics without modification, (**b**) G60-S40, and (**c**) G60-S40 with TEA.

**Figure 7 biomimetics-08-00198-f007:**
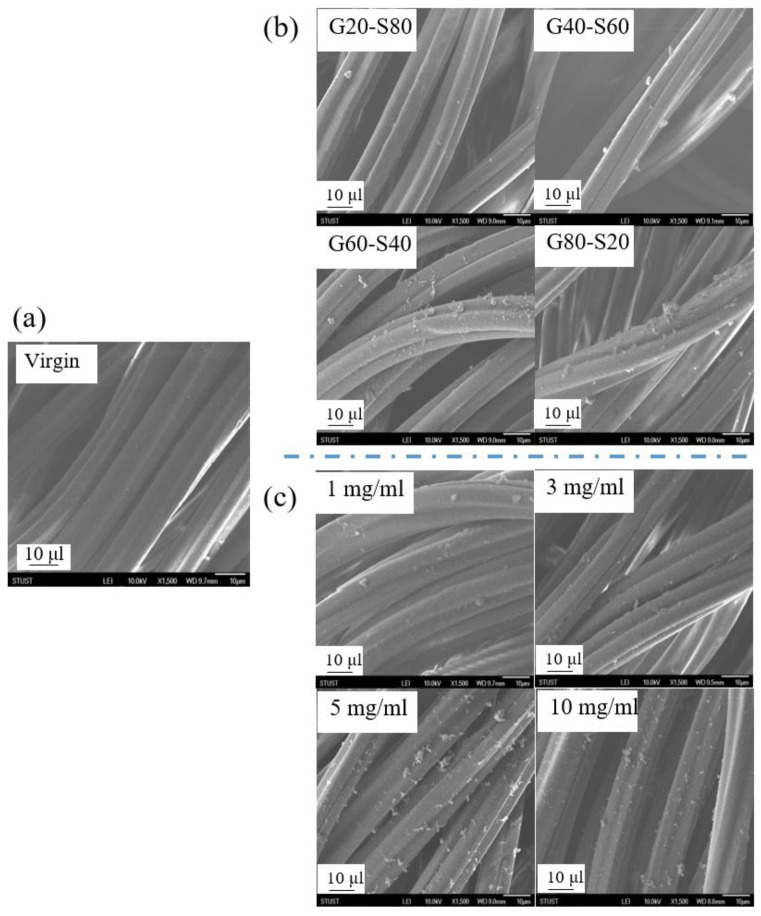
Observation of polymer morphology changes of polyamide elastic fabrics before and after (**a**) virgin unmodified substrate, (**b**) different molar ratios, and (**c**) different concentrations before and after modification by SEM.

**Figure 8 biomimetics-08-00198-f008:**
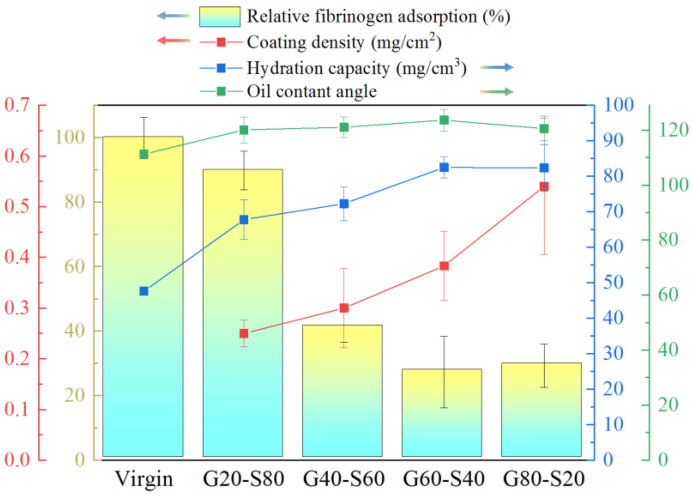
The relative changes of human fibrinogen adsorption amount and graft density, hydration force and oil contact angle.

**Figure 9 biomimetics-08-00198-f009:**
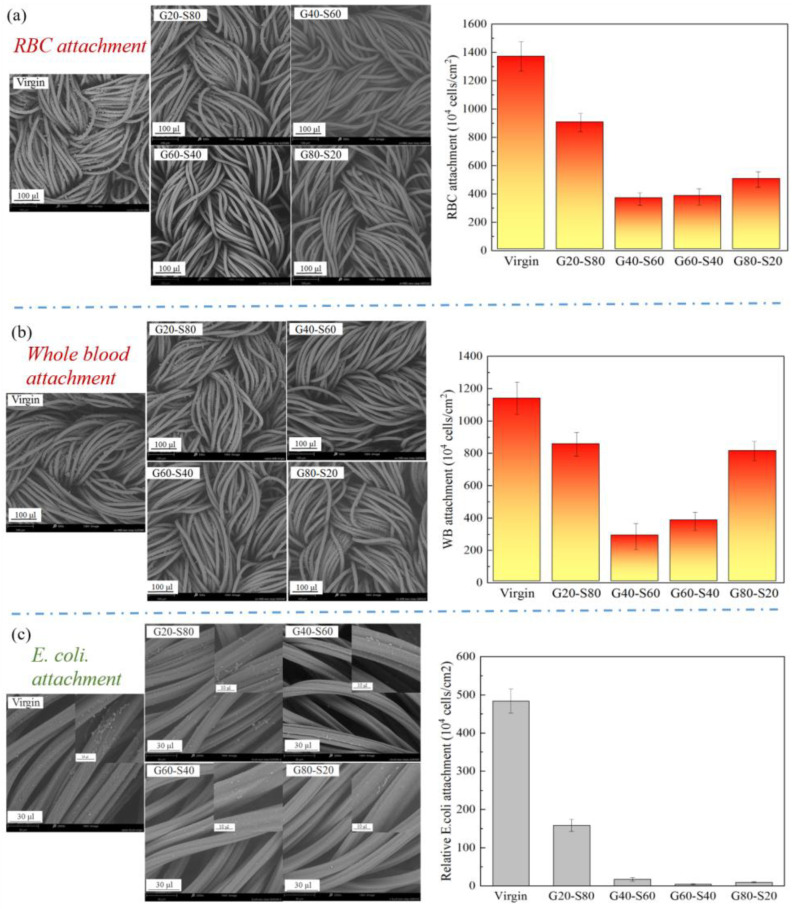
The amount of whole blood attached to the polyamide elastic fabric before and after modification (**a**) SEM image and statistics of the number of red blood cells attached, (**b**) SEM image and the statistics of the number of attached whole blood, and (**c**) SEM image and quantitative statistics of bacteria attachment.

**Table 1 biomimetics-08-00198-t001:** Poly(GMA-co-SBAA) polymer composition.

Sample ID	Reaction Ratios of Polymers (mol%) ^a^		Compositions of Polymers (mol%) ^b^		Characterization of Polymer	
	GMA	SBAA	GMA	SBAA	M_W_ (kDa)	PDI
SBAA	0	100	0	100	45.6	1.523
G10-S90	10	90	8	92	43.2	1.623
G20-S80	20	80	26	74	40.4	1.592
G30-S70	30	70	35	65	41.2	1.672
G40-S60	40	60	38	62	40.1	1.829
G50-S50	50	50	46	54	39.6	1.572
G60-S40	60	40	57	43	44.9	1.716
G70-S30	70	30	76	24	38.6	1.837
G80-S20	80	20	87	13	32.2	2.099
G90-S10	90	10	93	7	31.8	1.913
GMA	100	0	100	0	n/a	n/a

^a^ The molar ratio was indicated by the relative values between the monomers and initiator, with the mole of the initiator set at 1 mole compared to the monomers. ^b^ Composition of the poly(GMA-co-SBMA) copolymers were estimated by ^1^H NMR in D_2_O from the relative peak area of epoxide group of the pGMA in the range of δ at 4.55 ppm and that of (CH_3_)_2_N^+^ proton resonance of the pSBMA side groups in the range of δ at 3.2 and 3.4 ppm.

**Table 2 biomimetics-08-00198-t002:** Quantitative results of nitrogen and sulfur content in on the PA/PU fabrics.

Sample ID	Elemental Characterization of Copolymer Modified Surfaces *			
	%N	%N^+^	%S	N^+^:SO_3_^−^
Virgin	8.2	n/a	n/a	n/a
G60-S40	5.9	0.8542	0.9	0.95
G60-S40 with TEA	7.8	n/a	n/a	n/a

* The elemental ratios are calculated by the integration of XPS peak area and divide by the sum of all peaks in the survey scanning signal.

## Data Availability

Not applicable.

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
