# Peer review of "Biomimic Zwitterionic Polyamide/Polyurethane Elastic Blending Fabrics"

_biomimetics, 2023, doi:10.3390/biomimetics8020198_

Round 1
Reviewer 1 Report
Section 2 must be improved. Some proposed changes are the following:
- Minutes must be replaced by min in the whole text. The same for hours that must be replaced by h.
- Subsection 2.1. Reactants quality must be indicated, as well as the name of the supplier and country. Are the reactants as received or they need further purification?
- Subsection 2.3. should be improved by providing a more detailed description about the used experimental techniques and how the measurements were carried out. In all cases, provide more information about equipments.
- Contact angle measurements. Both polar and non polar liquids should be used in order to characterise hydrophilicity and hydrophobicity respectively.
- line 147. This sentence is incompleted.
- Most of the full stops are missing in the whole paper.
Author Response
Reviewer #1 – response
Comments and Suggestions for Authors
Section 2 must be improved. Some proposed changes are the following:
Remark 1. - Minutes must be replaced by min in the whole text. The same for hours that must be replaced by h.
Author Reply: Thanks for referee’s comment. We have corrected the whole article and replace “minute” with “min”.
Remark 2. Subsection 2.1. Reactants quality must be indicated, as well as the name of the supplier and country. Are the reactants as received or they need further purification?
Author Reply: Thanks for referee’s reminding. We have added a new section, 2.1, to describe the purchase and sources of materials. We also adjusted the numbering of the original session 2. The reactants used in this study were used directly without any further purification. From the observed results of the study, the reactants used directly were found to have sufficient reactivity, and thus no additional purification was necessary. For the reactant quality, we provide the reaction concentration and volume as shown in the following edited sentences.
“ Dissolve the monomers SBAA(sulfobetaine acrylamide) and GMA(glycidyl methacrylate) in 25 ml aqueous solution and 25 ml methanol solution, respectively, with a solid content of 20%, and add the two bottles of solutions into a 100 ml reaction bottle (Molar Ratio: SBAA:GMA:APS=40:60:1).”
Remark 3. Subsection 2.3. should be improved by providing a more detailed description about the used experimental techniques and how the measurements were carried out. In all cases, provide more information about equipments.
Author Reply: Thanks for referee’s reminding. We added the information of experimental equipment into each section. According to the suggestion, we have added more detailed descriptions to explain the experimental setup and methods. These are described in the new subsection 2.4 (formerly subsection 2.3) with the following paragraph:
“The types of functional groups on the surface of the material were identified using Fourier Transform Infrared Spectroscopy (FT-IR) (Perkin Ellmer Auto/MAGE FT-IR Microscope). The physical structure of the surface was observed using a scanning electron microscope (SEM, Phenom ProX, Thermo Fisher Scientific) at an accelerating voltage of 15KV. First, the sample was pasted on a special stage of a scanning electron microscope with carbon tape and put under vacuum condition for 24 hr. Before transfer into the SEM chamber, the sample was deposited with gold layer with 90-100 sec by the gold deposition machine to increase the electrical conductivity. X-ray photoelectron spectroscopy (PHI 5000 VersaProbe Ⅲ) measurements were conducted using a PHI Quantera SXM/Auger spectrometer with a monochromatic Al KR X-ray source (1486.6 eV photons), a hemispherical analyzer, and a multichannel detector. The elemental signal was determined from spectra acquired at pass energy form 50 eV to 150 eV. The detected photoelectron was acquired at the takeoff angle of 45 with respect to the sample surface. All the binding energy was referenced to the C 1s spectrum at 284.4 eV. In order to identify the polymer grafting density on the surface of the substrate, the polyamide elastic fabric was first cut into a square with a side length of 1 cm by an infrared cutting machine (SUII-4060, BAXCE technology, Taiwan), and put them into 24 Well plates, one piece per grid, and placed in the freezer. Remove the water on the surface in the drying machine, then take each piece of polyamide elastic fabric to a micro balance and weigh it (microbalancer AP135W, SHIMADZU) to obtain the weight before modification W1, then take each piece of polyamide elastic fabric to modify, and then remove the modified The finished polyamide elastic fabric is placed in a freeze dryer (FD-4.5-12P, Ker-shi tech.) to remove the water on the surface, and then weighed with a micro balance to obtain the modified weight W2.
Hydration capacity(mg/cm3)=(W2-W1)/V
W1: membrane weight before modification (mg)
W2: membrane weight after modification (mg)
V: membrane volume (cm3)
Because the high porosity and original hydrophilic property of polyamide fabrics, surface static or dynamic water contact angle can not sufficiently be detected. Thus, the surface hydrophilicity of polyamide elastic fabrics is measured by the oil contact angle by an angle-meter (FTA 125 Contact angle analyzer, USA). During the measurement, quantitative 4 μl of diiodomethane is dropped on the surface of the nylon fiber membrane and then measured by a contact angle meter The included angle between the droplet and the sample surface is used to judge the hydrophilicity of the samples.”
Remark 4. Contact angle measurements. Both polar and non polar liquids should be used in order to characterise hydrophilicity and hydrophobicity respectively.
Author Reply: Thanks for referee’s reminding. We should provide a more detailed description regarding the measurement of contact angle. As our substrate, polyamide elastic fabrics, is inherently porous and hydrophilic, polar liquids such as water are rapidly absorbed into the surface, making it difficult to measure contact angles accurately in both static and dynamic modes. Therefore, we used diiodomethane, a hydrophobic liquid, to represent the improvement in hydrophilicity (while oil contact angle increase). We also added the description in the new revision manuscript, line 224, as the following statements.
“Because the high porosity and original hydrophilic property of polyamide fabrics, surface static or dynamic water contact angle can not sufficiently be detected. Thus, the surface hydrophilicity of polyamide elastic fabrics is measured by the oil contact angle by an angle-meter (FTA 125 Contact angle analyzer, USA). During the measurement, quantitative 4 μl of diiodomethane is dropped on the surface of the nylon fiber membrane and then measured by a contact angle meter The included angle between the droplet and the sample surface is used to judge the hydrophilicity of the samples.”
Remark 5. line 147. This sentence is incompleted.
Author Reply: Thanks for referee’s reminding. We have revised this paragraph to improve its readability. The new sentences begin at line 160, as indicated in the following paragraph.
“Dissolve the monomers SBAA(sulfobetaine acrylamide) and GMA(glycidyl methacrylate) in 25 ml aqueous solution and 25 ml methanol solution, respectively, with a solid content of 20%, and add the two bottles of solutions into a 100 ml reaction bottle (Molar Ratio: SBAA:GMA:APS=40:60:1). After thoroughly stirring the solution, it should be filled with nitrogen gas for 10 minutes and placed in a silicon oil pot at 60°C for 6 hours. Then, the reaction bottle should be transferred to an ice bath at 4°C, and the solution should be poured into methanol for purification. The resulting polymers should be dried in a vacuum oven and then treated with a freeze dryer to remove any remaining water. Finally, the polymers should be stored in a vacuum-sealed container to ensure their stability. In this experiment, a nuclear magnetic resonance (NMR) spectrometer (Bruker, NMR-400MHz) with a resonant frequency of 400 MHz was used to detect the chemical structures of monomers and zwitterionic copolymers.”
Remark 6. Most of the full stops are missing in the whole paper.
Author Reply: Thanks for referee’s reminding. We re-check the whole article and confirm all the sentences are completed and the full stops are added.
------------------------------------------------
We hope that our answers and modifications will make our intentions and manuscript clearer. If editor and reviewers have any supplementary questions, comments or concerns, please do not hesitate to let us know.
Thank you very much.
Sincerely,
Ying-Nien Chou
Department of Chemical and Materials Engineering, Southern Taiwan University of Science and Technology, Tainan 71005, Taiwan

Reviewer 2 Report
Review report
The literature review is shallow, and there is a possible way to improve the research gap.
There are many typing errors (highlight, etc.) that should be removed.
Formaldehyde has been used for pre-treatment, it has severe environmental toxicities and issues., which the authors cannot justify.
Needs more explanation about Nylon 66; why are authors using it? Why not Nylon 6?
Clean NMR results are expected
Author Response
Reviewer #2 – response
Remark 1. - The literature review is shallow, and there is a possible way to improve the research gap.
Author Reply: Thanks for referee’s comment. In this revision, we have expressed the main purpose of this study more clearly. The study aims to develop a novel biomimetic zwitterionic copolymer that can effectively modify polyamide elastic fabrics. The results of various anti-biofouling experiments, including resistance to protein adsorption, bacterial attachment, and blood cell adhesion, have been promising. We believe that the technology developed in this study has broad and excellent applications in biomedical textiles.
Remark 2. There are many typing errors (highlight, etc.) that should be removed.
Author Reply: Thanks for referee’s reminding. We have reorganized the entire article and corrected errors.
We send this article for professional English editing by a native speaker. Especially, we also ask the editor to focus on the grammatical corrections for the experimental section. In this revision, we send the proofreading version and the final collection version of manuscript to show the English editing. More detailed revisions please refer to “2. Material and methods” in the new version of manuscript. The attached file includes the following parts.
Final collection manuscript
Proofreading version
Certificate of the editing “Paul Steed Certification of English Proofreading “
Remark 3. Formaldehyde has been used for pre-treatment, it has severe environmental toxicities and issues., which the authors cannot justify.
Author Reply: Thanks for referee’s comment. Indeed, formaldehyde has serious environmental toxicity, so any formaldehyde residue in biomedical materials is harmful. Therefore, in this study, only a slight amount of formaldehyde was used as a pretreatment to generate sufficient reactive hydroxyl groups. After formaldehyde treatment, the samples were subjected to multiple washes and modified with a biocompatible zwitterionic polymer in an alkaline environment. Generally, formaldehyde cannot maintain its reactivity (toxicity) after multiple washes and in subsequent reaction environments. Therefore, we regard that the formaldehyde used in this study as a pretreatment is acceptable for biomedical materials.
Remark 4. Needs more explanation about Nylon 66; why are authors using it? Why not Nylon 6?
Author Reply: Thanks for referee’s comment. Generally speaking, Nylon 66 is a higher grade material in textile applications due to its better strength and skin-friendliness. The polyamide elastic fabrics used in this study were provided by our partner, ITRI (Industrial Technology Research Institute, a textile research center), and are commonly used elastic fabrics in the industry. Therefore, the development of Nylon 66 can have better benefits for industrial applications. In addition, Nylon 66 and Nylon 6 have similar main structures in their chemical structures, so we believe that the related technologies developed in this study can be effectively used for both Nylon 66 and Nylon 6.
Remark 5. Clean NMR results are expected
Author Reply: Thanks for referee’s comment. As a polymer research article, we aimed to have NMR results with higher resolution. Unfortunately, despite our repeated efforts to purify and compare different NMR spectra, we have obtained similar results. The broad peaks observed in the polymer NMR of this study may be due to the formation of numerous hydrogen bonds along the amide-type copolymer backbone.
------------------------------------------------
We hope that our answers and modifications will make our intentions and manuscript clearer. If editor and reviewers have any supplementary questions, comments or concerns, please do not hesitate to let us know.
Thank you very much.
Sincerely,
Ying-Nien Chou
Department of Chemical and Materials Engineering, Southern Taiwan University of Science and Technology, Tainan 71005, Taiwan

Round 2
Reviewer 2 Report
Overall, the study on the “Biomimic zwitterionic polyamide/polyurethane elastic blending fabric”; however, there is no information found on the research gap, motivation for this work are the most selling point of the paper are missing. However, there are a few points that need to be addressed before it consider for the publications:
- Abstract should be rewritten and could be very crisp.
- Literature review is very shallow and most of the recent works are not cited properly.
- Figure 1 is not possible to read and either make them separate or better resolutions are required.
- Line 93, PA6 or PA66?
- Please mention the toxicity of the chemicals used for this work
Author Response
To Reviewer 2 (2nd response):
Thank you for the reviewers' suggestions. Through these suggestions, we have a better understanding of the direction of revision. We propose more revisions to improve this article.
Remark 1. - there is no information found on the research gap, motivation for this work are the most selling point of the paper are missing.
Author Reply: Thanks for referee’s comment. As suggested, we have added more literature discussion and explanation in the Introduction section, and provided a clearer introduction to the main selling points and features of this article. The modified content of the Introduction section is summarized as follows:
- In the third paragraph, we have added the latest relevant literature on zwitterionic copolymer materials and presented the current applications of epoxy-based zwitterionic copolymers. (new version line 82-89)
- In the last paragraph of the introduction, we have pointed out that the copolymer structure of poly(GMA-co-SBAA) has not been reported before and is a novel zwitterionic copolymer structure. Furthermore, through pre-treatment, sufficient hydroxyl groups are generated on the surface of the inert PA, which allows for effective ring-opening reactions and surface grafting modification, leading to surfaces resistant to biofouling. (new version line 128 and line 133)
- In the results and discussion, we provide an explanation for selecting the amide copolymer structure of poly(GMA-co-SBAA). This is mainly due to the amide structure of SBAA, which has a better modification effect on PA-PU elastic fibers. (new version line 298-303)
Remark 2. Abstract should be rewritten and could be very crisp.
Author Reply: Thanks for referee’s reminding. We have rewritten the abstract of the article to make it more concise and clear. It is shown below:
"This study developed an epoxy-type biomimic zwitterionic copolymer, poly(glycidyl methacrylate) (PGMA)-poly(sulfobetaine acrylamide) (SBAA), to modify the surface of polyamide elastic fabric using a hydroxylated pre-treatment zwitterionic copolymer and dip-coating method. X-ray photoelectron spectroscopy and Fourier transform infra-red spectroscopy confirmed successful grafting, while scanning electron microscopy revealed changes in surface pattern. Optimization of coating conditions included controlling reaction temperature, solid concentration, molar ratio, and base catalysis. The modified fabric exhibited good biocompatibility and anti-biofouling performance, as evidenced by contact angle measurements and evaluation of protein adsorption, blood cell and bacterial attachment. This simple, cost-effective zwitterionic modification technology has high commercial value and is a promising approach for surface modification of biomedical materials."
Remark 3. Literature review is very shallow and most of the recent works are not cited properly.
Author Reply: Thanks for referee’s comment. We revised the content of the introduction section and attempted to cite recent relevant studies. The added citation is included in the text of the introduction section as follows: (new version line 82-89)
- 17. Tang, S.H.; Domino, M.Y.; Venault, A.; Lin, H.T.; Hsieh, C.; Higuchi, A.; Chinnathambi, A.; Alharbi, S.A.; Tayo, L.L.; Chang, Y.Bioinert Control of Zwitterionic Poly(ethylene terephtalate) Fibrous Membranes. Langmuir 2019, 35, 1727–1739, doi:10.1021/acs.langmuir.8b00634.
- Chou, Y.-N.; Venault, A.; Cho, C.-H.; Sin, M.-C.; Yeh, L.-C.; Jhong, J.-F.; Chinnathambi, A.; Chang, Y.; Chang, Y.Epoxylated Zwitterionic Triblock Copolymers Grafted onto Metallic Surfaces for General Biofouling Mitigation. Langmuir 2017, 33, 9822–9835, doi:10.1021/acs.langmuir.7b02164.
- Lin, H.T.; Venault, A.; Chang, Y.Zwitterionized chitosan based soft membranes for diabetic wound healing. J. Memb. Sci. 2019, 591, 117319, doi:10.1016/j.memsci.2019.117319.
- Lien, C.C.; Chen, P.J.; Venault, A.; Tang, S.H.; Fu, Y.; Dizon, G.V.; Aimar, P.; Chang, Y.A zwitterionic interpenetrating network for improving the blood compatibility of polypropylene membranes applied to leukodepletion. J. Memb. Sci. 2019, 584, 148–160, doi:10.1016/j.memsci.2019.04.056.
- Dizon, G.V.; Clarin, M.T.R.; Venault, A.; Tayo, L.; Chiang, H.C.; Zheng, J.; Aimar, P.; Chang, Y.A Nondestructive Surface Zwitterionization of Polydimethylsiloxane for the Improved Human Blood-inert Properties. ACS Appl. Bio Mater. 2019, 2, 39–48, doi:10.1021/acsabm.8b00212.
- Fowler, P.M.P.T.; Dizon, G.V.; Tayo, L.L.; Caparanga, A.R.; Huang, J.; Zheng, J.; Aimar, P.; Chang, Y.Surface Zwitterionization of Expanded Poly(tetrafluoroethylene) via Dopamine-Assisted Consecutive Immersion Coating. ACS Appl. Mater. Interfaces 2020, 12, 41000–41010, doi:10.1021/acsami.0c09073.
Remark 4. Figure 1 is not possible to read and either make them separate or better resolutions are required.
Author Reply: Thanks for referee’s comment. We have redesigned Figure 1 to present the process in a clearer and more straightforward top-down flowchart format. Additionally, we have increased the resolution of the objects in the image, as shown below:
Figure 1. Schematic illustration of the zwitterionic modification of PA/PU fabrics through formaldehyde pretreatment and ethoxylated poly(GMA-co-SBAA).
Remark 5. Line 93, PA6 or PA66?
Author Reply: Thanks for referee’s comment. The polyamide discussed in line 93 (in new version change to line 90-92) come from the origin reference [16]: “In vitro studies of platelet adhesion on UV radiation-treated nylon surface”. We check this reference and they did not mention whether it is PA6 or PA66. Generally, PA66 exhibits higher tensile strength, rigidity, and hardness compared to PA6. Thus, it’s majorly applied in textile industry as high-value fabrics, normally applied in high-price textiles. However, PA6 tends to have better impact resistance and is more easily processed, therefore many of engineering plastics are made of PA6.
Remark 6. Please mention the toxicity of the chemicals used for this work
Author Reply: Thanks for referee’s reminding. We added a section on the toxicity of formaldehyde in the experimental part, emphasizing the need for safe handling precautions (new version line 184-186). In addition, we added a paragraph on formaldehyde toxicity in the results and discussion section, stating that formaldehyde is only used to produce hydroxyl groups and must be thoroughly washed and treated to avoid residual toxicity. (new version line 361-364)
------------------------------------------------
We hope that our answers and modifications will make our intentions and manuscript clearer. If editor and reviewers have any supplementary questions, comments or concerns, please do not hesitate to let us know.
Thank you very much.
Sincerely,
Ying-Nien Chou
Department of Chemical and Materials Engineering, Southern Taiwan University of Science and Technology, Tainan 71005, Taiwan
